# End-of-Life Care during the COVID-19 Pandemic: Decreased Hospitalization of Nursing Home Residents at the End of Life

**DOI:** 10.3390/healthcare12161573

**Published:** 2024-08-08

**Authors:** Helena Bárrios, José Pedro Lopes Nunes, João Paulo Araújo Teixeira, Guilhermina Rêgo

**Affiliations:** 1Hospital do Mar Cuidados Especializados Lisboa, 2695-458 Bobadela, Portugal; 2Faculty of Medicine, University of Porto, 4099-002 Porto, Portugal; jplnunes@med.up.pt (J.P.L.N.); jpat@med.up.pt (J.P.A.T.)

**Keywords:** end-of-life care, care transitions, nursing home, palliative care, geriatrics, COVID-19

## Abstract

(1) Background: Nursing homes (NHs) face unique challenges in end-of-life care for their residents. High rates of hospitalization at the end of life are frequent, often for preventable conditions. The increased clinical uncertainty during the pandemic, the high symptom burden of the COVID-19 disease, and the challenges in communication with families and between care teams might impact the option to hospitalize NH residents at the end of life. (2) Materials and methods: The study covered a 3-year period and compared the hospitalization rates of the NH residents of a sample of Portuguese NH during the last year of life before and during the pandemic. A total of 387 deceased residents were included in the study. (3) Results: There were fewer hospitalizations in the last year of life during the pandemic period, although the proportion of deaths at hospitals was the same. Hospitalizations occurred closer to death, and with more serious clinical states. The lower rate of hospitalization was due to lower hospitalization due to infection; (4) Conclusions: The data suggest an improvement in end-of-life care practices during the pandemic period, with the decrease in hospitalizations being due to potentially burdensome hospitalizations. The importance of the role of physicians, nurses, and caregivers in this setting may be relatively independent of each other, and each may be targeted in end-of-life care training. Further study is recommended to clarify the implications of the results and if the changes can be sustained in the long term.

## 1. Introduction

Nursing home (NH) residents are among the frailest group of older people [1,2,3,4]. The last years of life in this group are often characterized by clinical uncertainty over recovery or continued deterioration leading to death [5]. End-of-life care requires parallel planning and intervention to support recovery and anticipate and plan for deterioration and dying [6].

The NH setting presents unique challenges in dealing with the complexity of end-of-life care. The limitations of low clinical support, high levels of uncertainty related to residents’ chronic clinical status, frequent acute health events, and conflicting interests and beliefs among residents, their families, and NH staff contribute to difficulties in maintaining a consistent and effective end-of-life care plan [6,7,8,9].

The COVID-19 pandemic posed unprecedented challenges to the care of NH residents. It was caused by a new coronavirus (2019-nCoV), first identified as causing an acute respiratory syndrome in humans in November 2019, in Wuhan, China [10]. The disease rapidly evolved into a pandemic declared by the World Health Organization in March 2020 [11]. Knowledge of the pathogenesis, clinical expression, and adequate preventive and treatment measures was built during the ongoing pandemic. Amid scientific uncertainty, WHO, and country guidance and regulations have often been contradictory in the early stages of the pandemic. Moreover, personal protective equipment, testing, and later vaccination have suffered severe shortages [11,12,13,14]. In this context, NH suffered a particularly high toll during the pandemic. Given its organization, architectural characteristics (closed systems, proximity between users and staff), and susceptible population (age, comorbidities, specific care needs) [15], NHs are prone to high transmissions of COVID-19 disease, with high morbidity and mortality rates being described worldwide [16,17,18,19,20].

End-of-life care in NHs was even more challenging during the COVID-19 pandemic. There was a need for new protocols, the symptom management could be complex due to the nature of the COVID-19 disease, social distancing increased the difficulty of discussing decisions with family members and loved ones, and the articulation between the NH and Hospital was hindered by restrictions to scheduled activity and overcrowding of emergency departments [21,22,23].

Emergency department referral is a frequent option to manage deteriorating health conditions in NH residents. Hospital evaluation can be useful in reversing acute conditions, stabilizing decompensations of chronic illnesses, or evaluating and treating trauma consequences [24]. However, in frail patients, at the end of life, hospitalization can be burdensome to the patient and might be considered a marker of healthcare aggressiveness [25,26]. In Portugal, emergency departments are chronically overused by the general population. Moreover, in Portugal, NH residents are frequently referred to the emergency department in the last year of life, and death at the hospital is frequent, both in the general population and NH residents [27,28].

During the pandemic, a decrease in hospital admissions was identified both in the general population [29,30] and in NH residents [31,32,33]. However, in these studies, it is not clear if the residents at the end of life also had fewer hospitalizations during the pandemic. Given the complex nature and symptoms of the COVID-19 disease, it might be expected that higher rates of hospitalization occur in these patients near the end of life. Moreover, the previously described difficulties over end-of-life discussions with family members and caregivers, given the restrictions imposed by the pandemic, could negatively impact the decision-making process regarding hospitalization. These factors could increase the hospital admissions of NH residents at the end of life, in contrast with the previously reported marked decrease in hospital admissions of the general NH resident population. In the Portuguese context, a decrease in general hospital admissions has been described but the NH resident has not been previously studied [34]. With the present study, the authors aim to identify the impact of the COVID-19 pandemic on NH resident hospitalization at the end of life, place of death, and deceased NH resident characteristics.

## 2. Materials and Methods

The study was designed to address the research questions: (1) What was the impact, if any, of the COVID-19 pandemic on NH resident hospitalization at the end of life and place of death? (2) What were the differences, if any, in the characteristics of deceased NH residents during the same pandemic, when compared to the year and a half before the pandemic?

### 2.1. Study Design

We conducted a cohort retrospective after-death study. The present study covered a 3-year period, from September 2018 to August 2021. This time frame included 18 months during the pandemic period and 18 months prior to its beginning (March 2020 in Portugal) [35]. STROBE criteria were followed in the design and report of the present study [36].

#### Study Population

The present study followed a previous study conducted in a sample of Portuguese NH residents that aimed to identify and characterize acute care transitions in the last year of life [28]. In the previous study, a convenience sample, stratified by geographical region, of 7 NHs, harboring a total of 614 residents was selected from a total of 387 NH charities affiliated with União das Misericórdias Portuguesas. A convenience sample was chosen to allow the feasibility of the study. Stratification by region involved selecting a proportional number of participants in each Portuguese region in order to improve the representability of the data. Data were initially collected for a sample of 176 deceased residents in the 18 months prior to the COVID-19 pandemic (minimal sample size calculated at 138 subjects). For the present study, additional data were collected for the residents who died in the first 18 months of the pandemic. All deaths that occurred within the study period were analyzed. Death in the 90 days after admission was considered exclusion criteria, since the discussion of advanced directives and the relationship with the resident and their family may influence the decision to transfer the resident to acute care and the place of death. A 90-day period to allow for these discussions to take place is recommended in the literature [37,38].

### 2.2. Variables and Instruments

Variables related to the resident NH healthcare team and the hospitalization episodes were studied. The variables related to the resident were: age at the time of death; gender; length of stay in the NH; chronic health diagnosis; cause of death; place of death; performance status on the month prior to death, using Eastern Cooperative Oncology Group (ECOG)—Performance Status Scale (PS). The ECOG Performance Status Scale is a widely used method to assess the functional status of a patient. The instrument has interrater reliability described in the literature from 0.69 to 0.83 (Spearmen rank correlation) [39]. It is composed of six categories: 0: Fully active, able to carry on all pre-disease performance without restriction; 1: Restricted in physically strenuous activity but ambulatory and able to carry out work of a light or sedentary nature, e.g., light housework, office work; 2: Ambulatory and capable of all self-care but unable to carry out any work activities, up and about more than 50% of waking hours; 3: Capable of only limited self-care, confined to bed or chair more than 50% of waking hours; 4: Completely disabled, cannot carry on any self-care, totally confined to bed or chair; 5: Dead [40]. A questionnaire covering the characteristics of the NH (number of residents, total of deaths in the institution in the study period, NH care team constitution, and professional/caregivers’ ratio) was previously sent to the participating NH. The data were collected in an interview conducted by the principal investigator with the NH management team. In the present study, physician, nurse, and caregiver ratios were analyzed. NH teams in the institutions enrolled also include social workers and therapists; however, these professionals are not included in clinical decisions regarding hospitalizations and, therefore, have not been included in the study. Clinical pharmacists, do not usually integrate the NH in Portugal and were not a part of the NH care team in the studied institutions; therefore, they have not been included in the study. The constitution of the care team remained stable before and during the pandemic. No external team support was provided during the pandemic to the NH enrolled in the study. Data on hospitalizations during the 12 months prior to death were also studied: number of hospital admissions; days spent at the hospital; motif for referral and discharge diagnosis. Data were collected through analyses of the administrative and clinical records of the deceased residents, death certificates, and hospital discharge information.

### 2.3. Statistical Analysis

Sample characterization was performed using descriptive analyses. To compare the characteristics of the deceased residents and hospitalizations before and after the pandemic, the t-student test for 2 independent samples was used for continuous variables, as well as for *χ*^2^ for dichotomous variables. The correlation between team ratios and hospitalizations was studied using the Pearson correlation coefficient. Cases were excluded pair-wise in the case of missing data. All reported *p*-values are two-sided with significance defined as *p* < 0.05. All analyses were performed with SPSS 28.01.0 for Windows (IBM Corporation, Armonk, New York, NY, USA).

## 3. Results

### 3.1. Mortality and Place of Death

A total of 454 NH deceased residents were screened for eligibility. A total of 67 were excluded due to death within the 90 days following admission. Therefore, 387 deceased residents were included in the study (Table 1), 176 before the pandemic, and 211 in the pandemic period (*χ*^2^ (1, N = 1228) = 4.62, *p* = 0.03). 58 deaths (28.8% of the deaths during the pandemic) were due to COVID-19 infection. During the pandemic period, 153 residents died due to non-COVID-19 reasons, a non-significant difference from the pre-pandemic mortality (*χ*^2^ (1, N = 1228) = 2.20, *p* = 0.14). The difference between the proportion of deaths at hospitals before and during the pandemic period was not statistically significant (*χ*^2^ (1, N = 387) = 0.34, *p* = 0.56), even if COVID-19 deaths are excluded (*χ*^2^ (1, N = 329) = 0.88, *p* = 0.35).

### 3.2. Deceased Resident Characteristics

The total sample studied was composed mostly of women (66.7%); residents died aged 86.1 ± 8.7 years, lived for more than 4 years at the NH (54.3 ± 53.7 months), had high levels of disability (58.1% PS 4), multimorbidity (3.3 ± 1.4 diagnoses), and 83.7% had a diagnosis of cognitive impairment.

Statistically significant differences in the characteristics of deceased NH residents before and during the pandemic period were found for PS, number of co-morbidities, and cause of death. The included NH residents had a PS ranging from 2 to 4. Prior to the pandemic period, the large majority had a PS of 4 (69.9%), and only 4.0% had a PS of 2. During the pandemic period, deceased residents had better PS, with 10.4% of deceased residents with PS 2 (*χ*^2^ (1, N = 387) = 5.76, *p* = 0.02), 48.3% with a PS 4 (*χ*^2^ (1, N = 387) = 18.30, *p* < 0.001). If residents who died due to COVID-19 infection were excluded from the analysis, a statistically significant difference persists for PS3 (*χ*^2^ (1, N = 330) = 7.22, *p* = 0.007) and PS4 (*χ*^2^ (1, N = 330) = 10.22, *p* = 0.001). Moreover, fewer comorbidities were found in NH residents dying during the pandemic period (t(385) = 2.13, *p* = 0.03), however, this difference loses significance if residents who died due to COVID-19 are excluded (t(322) = 1.46, *p* = 0.08). The number of deaths from infection excluding COVID-19 is significantly lower during the pandemic period (45.5% in the pre-pandemic period, 21.4% during the pandemic (*χ*^2^ (1, N = 376) = 27.83, *p* < 0.001). Analyzing COVID-19 and non-COVID-19 infection together, the difference in the proportion of deaths from infection is not statistically significant (*χ*^2^ (1, N = 376) = 0.23, *p* = 0.64).

We found no statistically significant difference in the other variables under study namely, gender distribution, age at time of death, length of stay in the NH, prevalence of specific medical conditions (dementia, arterial hypertension, heart disease, diabetes mellitus, stroke, chronic renal disease, lung disease), death from cardiac or neurological causes or cancer.

### 3.3. Hospitalization in the Last Year of Life

During the pandemic period, there were fewer hospitalizations of residents in the last year of life: 327 in the pre-pandemic period vs. 191 during the pandemic (*p* < 0.001) (Table 2). During the pandemic, 28% of the deceased residents were never transferred to hospital in the last year of life (significantly more than in the pre-pandemic period (17.5%) (*χ*^2^ (1, N = 387) = 5.76, *p* = 0.02). If transferals due to COVID-19 infection are excluded, the number is even more expressive, with 43.1% of the residents never being hospitalized in the last year of life for non-COVID-19 diagnosis (*χ*^2^ (1, N = 387) = 28.94, *p* < 0.001). When hospitalization occurred, most of the residents were transferred once, unlike before the pandemic, when the majority had two or more hospitalizations (*χ*^2^ (1, N = 387) = 17.59, *p* =< 0.001). The number of nights per episode of hospitalization was similar before and during the pandemic period (t(516) = 0.18, *p* = 0.43). However, given the lower number of hospitalizations, the total number of nights spent at the hospital per resident during the last year of life was lower during the pandemic (t(384) = 2.33, *p* = 0.01). In the pandemic period, residents were hospitalized closer to death, with more hospitalizations occurring 1 to 30 days prior to death: *χ*^2^ (1, N = 518) = 19.27, *p* =< 0.001 and *χ*^2^ (1, N = 518) = 7.53, *p* = 0.006), and less with hospitalizations more than 90 days prior to death (*χ*^2^ (1, N = 518) = 37.86, *p* =< 0.001). If only non-COVID-19 hospitalizations are included, the difference remains significant with more hospitalizations 1 to 3 days before death (*χ*^2^ (1, N = 471) = 5.95, *p* = 0.02) and fewer hospitalizations more than 90 days prior to death (*χ*^2^ (1, N = 471) = 19.38, *p* < 0.001). Considering the motifs for referral, and excluding the hospitalizations due to COVID-19 disease, more residents were transferred due to altered state consciousness during the pandemic (*χ*^2^ (1, N = 471) = 12.14, *p* < 0.001) (Table 3). No statistically significant difference was found on transferals due to falls, bleeding, dyspnea or other respiratory symptoms, and focal neurological symptoms. Regarding discharge diagnosis, fewer residents had a diagnosis of infection (if COVID-19 infection is excluded) (*χ*^2^ (1, N = 518) = 7.72, *p* = 0.006). Neurological, cardiac disease, cancer-related hospitalizations, or trauma had no statistically significant difference before and after the beginning of the pandemic. 60.7% of hospitalizations ended with the death of the resident, a significantly higher rate than before the pandemic, 28.1% (*χ*^2^ (1, N = 515) = 36.02, *p* < 0.001).

### 3.4. NH Health Team Constitution and Hospitalization in the Last Year of Life

In the present sample, a significant positive correlation was found between the nurse/resident ratio and resident hospitalization during the last year of life during the pre-pandemic period (a higher ratio, e.g., more nurse support is correlated with more hospital referrals) (Table 4). This correlation is maintained only for hospitalizations in the last 30 days of life during the COVID-19 pandemic. Additionally, the caregiver/resident ratio was also positively correlated with hospitalization in the 30 days of life in the pre-pandemic period, but not for the COVID-19 pandemic period. No correlation was found between physician/resident ratio and hospitalization.

## 4. Discussion

The COVID-19 pandemic was a unique event, one that involved high levels of uncertainty and led to high morbidity and mortality [41,42]. In line with NHs worldwide [12], NHs in the present sample were significantly affected by the pandemic, with an increase in the number of deaths above what would normally be expected based on the experience from previous years, and 29% of the deaths in this period being due to COVID-19 disease.

Despite the seriousness of the COVID-19 disease and the social alarm associated with it, the hospitalizations of NH residents in the last year of life significantly decreased in relation to the pre-pandemic period, a finding even more expressive if COVID-19-related transferals are excluded. The proportion of residents who were never hospitalized increased from 17.5% pre-pandemic to 43.1% in the pandemic period (excluding transferals due to COVID-19 disease). When hospitalization did occur, most of the residents were transferred once, unlike before the pandemic, when the majority had two or more hospitalizations in the last year of life.

Previous studies analyzing the global hospital admissions [29] and hospitalizations of NH residents have described this trend [31,32,33,43], although hospitalization in the last year of life has not been previously studied independently. Several factors may have contributed to this result. In Portugal, infection control measures implemented by the national authorities imposed isolation periods in the NH after hospitalization [44]; this drawback, associated with the fear of contracting COVID-19 in the hospital, may have discouraged emergency department referral by the NH team [33]. An increased threshold for emergency care transferal, with longer periods of treatment and observation in the NH, might have contributed to a decrease in inappropriate referrals [31]. Although the operationalization of the concept of inappropriate hospital use may be questioned [45], there is a wide consensus that in NH residents with advanced illness, hospitalization near the end of life is potentially avoidable if the acute condition can be managed effectively in the nursing home or the hospital-level care is not aligned with patient preferences [46,47]. In our sample, the decrease in hospitalization was due to a decrease in the hospitalization due to infection, a widely accepted cause of preventable hospital referral [46,47,48,49]. No statistically significant difference was found in transferals due to other causes. In some studies, an increase in healthcare support to the NH by external teams was described as contributing to decreased hospitalizations [32], a factor not identified in the present sample. The decrease in hospitalization at the end of life in frail NH residents, independently of its’ cause, may be considered an improvement in end-of-life care practices [50].

In the pandemic period, residents were hospitalized closer to death. The lower number of admissions was mainly due to fewer hospitalizations more than 90 days prior to death, favoring the interpretation that the reduced number of admissions was due to preventable causes. The proportion of hospitalizations in the last days of life and with altered states of consciousness increased. Increased delirium at referral and hospital use of antipsychotic drugs have been described in a Canadian study [32] and attributed to the higher social isolation of the residents. However, the hypothesis of transferal at a more serious and advanced disease condition, with delirium, and a higher need for symptom control medication should not be overruled.

Death from infectious causes other than COVID-19 decreased significantly during the pandemic period. This phenomenon has been described in other studies, with the lower mortality from other transmissible diseases being attributed to measures taken to mitigate the spread of COVID-19 infection, namely, stricter hygiene standards, regular disinfection, social distancing, and screening of visitors [13,15]. Death from noninfectious causes remained stable during the pandemic.

The proportion of NH residents dying at hospitals did not change significantly during the pandemic period. The rate of death at the Hospital of NH residents has high variability between studies [25]. Part of the variability is explained by cultural issues, with different countries having distinct profiles of choice of place of death. In Portugal, the choice of the hospital as the favored place of death has been previously described, with high rates of death at hospitals of NH residents (53%) [28] and the general population (66.7%) [27]. The fact that the proportion of hospital deaths did not increase in the face of higher uncertainty may be seen as positive in this context.

In this sample, excess mortality was due directly to COVID-19 disease. Excess mortality was identified worldwide [41,51,52], in all levels of care [53,54], although frail elderly living at NHs were particularly affected [20,55,56]. Excess mortality related to COVID-19 may have been caused either directly by COVID-19 disease or indirectly, for example, due to healthcare system overload [52,53]. The proportion of excess mortality attributed directly to COVID-19 is open to some debate due to variable levels of underreporting, more significant in some settings and countries than others, namely, home deaths and countries with less strict and available diagnoses of COVID-19 infection [41,54]. In Portugal, a widespread proactive testing policy implemented in NH might have minimized the underreport of COVID-19 infection in NH and contributed to the absence of significant excess mortality due to other causes identified in this sample [35,57].

Although the mortality globally increased during the pandemic period, differences were minor in the socio-demographic and clinical characteristics of the NH residents that died during the study period, indicating homogeneity in the population even in the very serious event of a pandemic. More residents presented with PS 3 and 2 in the month prior to the death pandemic period (a difference that only prevails for PS3 if COVID-19 deaths are excluded). However, the apparently better PS in the residents dying during the pandemic period should be interpreted with caution since the residents have an altogether high disability with more than 90% having PS 3 (confined to bed or chair more than 50% of waking hours) or PS4 (totally confined to bed or chair). This might indicate that the population is very dependent, frail, and with a high prevalence of comorbidities, with a ceiling effect being reached before the pandemic.

Unlike other international studies [58,59], in this sample, higher nurse team support (and in the last 30 days of life, higher caregiver support) was correlated with higher hospitalizations, in the pre-pandemic period. These findings have been previously described [28] and attributed to the difficulty of the NH team in dealing with end-of-life issues in a country with low palliative care awareness that favors the hospital as the appropriate place to die [27,28,60]. The fact that, during the COVID-19 pandemic, there was decreased referral to acute care and that the correlation with the NH healthcare team was identified only for the nursing team in the last 30 days of life, highlights the influence that this team has on clinical transferal decisions. This supports the need to improve end-of-life and palliative care training for NH teams, particularly the nursing teams [58].

In Portugal, NH remains officially a social care response to the elderly. The only mandatory healthcare professional is a nurse for a limited period of the day. Even so, nowadays, indoor medical support is the norm. However other healthcare professionals like clinical pharmacists or nutritionists are very rarely included in the care teams of the NH [61]. Clinical pharmacists’ intervention in NH has been proven to be effective in ensuring quality use of medication, resulting in reduced fall rates [62,63,64]. Although evidence is less clear on the impact of reducing mortality or hospitalization and admission rates, filling this gap in the Portuguese NH might help to improve outcomes in end-of-life care [63].

### 4.1. Limitations

Being an after-death study, the study design is retrospective. This is a limitation common to other mortality studies on the hospitalization of NH residents at the end of life [25,59,65]. To minimize data loss, information on sociodemographic characteristics and hospital referrals was obtained by linking administrative and clinical data, and death diagnoses were obtained from the national electronic death certificate database. Given the retrospective analysis of the last year of life, there is an overlap in the analysis of pre-pandemic hospitalizations in residents who died during the first year of the pandemic period. This fact would tend to decrease differences between the two groups. However, the results obtained evidenced a significant decrease in hospitalization, and an increase in hospital admissions near death, a trend opposed to the bias introduced, that may contribute to its mitigation. Although the sample size is small, it represents a sample of a large subset of Portuguese NH, and the included NH has a nationwide distribution. The information on the healthcare team constitution was obtained through interviews, and not formal NH document analysis, which may introduce some bias.

### 4.2. Future Prospects

The results obtained can be interpreted as an improvement of end-of-life care practices in the NH during the pandemic. The NH teams were able to provide adequate care to residents with infection in the last year of life and avoid potentially burdensome hospitalizations [46]. Kasdorf and co. described the most relevant aspects that can aid in reducing unnecessary transitions: timely identification and communication of the last year of life; consideration of palliative care options; availability and accessibility of care services; and having a healthcare professional taking main responsibility for care planning [7]. During the pandemic public discussion on ceilings of care and identification of prognosis factors might have contributed to the awareness, both by the care team and the residents and their families, about the need to provide proportional care to frail elderly patients [22,66]. During the COVID-19 pandemic, the multidisciplinary care team at the NH, particularly the nursing team was able to better identify the needs of the residents and its own capacities for symptom control, and optimize care delivery, using a more proportional approach. Targeting end-of-life training and fostering multidisciplinary care team practices at the NH may contribute to improving and maintaining better end-of-life care practices after the COVID-19 pandemic [67].

## 5. Conclusions

This study brings new insights relating to hospitalization at the end of life of NH residents during the COVID-19 pandemic. A significant decrease in hospitalizations was identified during the pandemic, although mortality increased (with excess mortality attributable to COVID-19 disease), and the rate of death at the hospital remained stable. More residents died without visiting the emergency room in the last year of life and spent fewer nights at the hospital. When hospitalization did occur, it was closer to death and with more serious clinical presentations. The decrease in hospitalizations was due mostly to potentially burdensome admissions, namely, non-COVID-19 infections.

The deceased NH resident characteristics were similar in the period before and during the pandemic. Residents were very old, with high levels of dependency and multi-morbidity, with a possible ceiling effect reached before the pandemic.

A decrease in referrals and less burdensome transitions suggest an improvement in end-of-life care practices during the pandemic period. This indicates that NH care teams were able to successfully meet some of the challenges of end-of-life care facing a devastating pandemic. The data suggest that the influence of the role of physicians, nurses, and caregivers in the hospitalization of NH residents may be relatively independent of each other. The nursing team was revealed to be particularly influential on transferal decisions. Improved training in end-of-life care targeting the care team, and prioritizing the nursing team, may contribute to sustaining the positive changes described in the future. Further qualitative studies may highlight other factors contributing to this result and inform good practices that can be sustained in the future.

## Figures and Tables

**Table 1 healthcare-12-01573-t001:** Comparison of socio-demographic and clinical characteristics of deceased NH residents. In the period before and during the pandemic. N—number; SD—standard deviation.

Variable	Prepandemic	Pandemic	Pandemic (Excluding COVID-19 Deaths)
Residents, N	176	211 *	153
Gender, N (%)			
Male	53 (30.1)	76 (36.0)	56 (36.6)
Female	123 (69.9)	135 (64.0)	97 (63.4)
Age at death, years mean ± SD	85.6 ± 8.6	86.4 ± 8.7	86.1 ± 8.8
Length of stay, months mean ± SD	52.6 ± 46.0	55.8 ± 59.5	57.5 ± 61.5
Performance status, N (%)			
2	7 (4.0)	22 (10.4) *	11 (7.2)
3	46 (26.1)	87 (41.2) **	61 (39.9) **
4	123 (69.9)	102 (48.3) ***	81 (52.9) ***
Number of chronic diseases, N mean ± SD	3.5 ± 1.5	3.15 ± 1.4 *	3.2 ± 1.4
Chronic diseases, N (%)			
Dementia	153 (86.9)	171 (81.0)	126 (82.4)
Arterial hypertension	121 (68.8)	136 (64.5)	95 (62.1)
Heart disease	76 (43.2)	74 (35.1)	55 (35.9)
Diabetes mellitus	56 (31.8)	57 (27.0)	40 (26.1)
Stroke	41 (23.3)	45 (21.3)	33 (21.6)
Chronic renal disease	32 (18.2)	26 (12.3)	20 (13.1)
Lung disease	24 (13.6)	32 (15.2)	24 (15.7)
Cancer	24 (13.6)	19 (9.0)	16 (10.5)
Cause of death, N (%)			
Infection	80 (45.5)	102 (49.6)	
*Non-COVID-19*		*44 (21.4) ***	
*COVID-19*		*58 (28.2)*	
Neurological	29 (16.5)	40 (19.4)	
Cardiac	21 (11.9)	15 (7.3)	
Cancer	15 (8.5)	9 (4.4)	
Other	31 (17.6)	40 (19.4)	
Death at hospital, N (%)	92 (52.3)	116 (55.2)	72 (47.4)

* *p* < 0.05; ** *p* < 0.01; *** *p* < 0.001.

**Table 2 healthcare-12-01573-t002:** Comparison of acute care referrals in the last year of life, per resident. N—number, SD—Standard deviation.

Variable	Death in the Prepandemic Period	Death in Pandemic Period	Death in the Pandemic Period (Excluding COVID-19 Deaths)
Referrals to acute care in the last year of life, per resident, mean ± SD	1.6 (1.4)	1.1 (1.0) ***	0.9 (1.0) ***
Referrals to acute care in the last year of life, per resident, N (%)			
0	31(17.5)	59 (28.0) *	91 (43.1) ***
1	65 (36.7)	99 (46.9) *	72 (34.1)
≥2	80 (45.5)	53 (25.1) ***	48 (22.7) ***
Nights at the Hospital, mean ± SD	7.9 (14.3)	4.9 (9.9) *	4.3 (9.9) **

* *p* < 0.05; ** *p* < 0.01; *** *p* < 0.001.

**Table 3 healthcare-12-01573-t003:** Comparison of acute care referrals in the last year of life, per referral episode. N—number.

Variable	Prepandemic	Pandemic	Pandemic (Excluding COVID-19 Deaths)
Referrals to acute care in the last year of life, per episode, N	327	191 ***	144 ***
Number of nights at the hospital per episode, mean ± SD	4.5 ± 7.6	4.4 ± 8.6	4.6 ± 9.6
Days prior to death			
On day of death	26 (9.3)	22 (11.5)	18 (12.5)
1–3 days	38 (11.6)	51 (26.7) ***	29 (20.1) *
4–30 days	69 (21.1)	61 (31.9) **	40 (27.8)
31–90 days	57 (17.4)	27 (14.1)	27 (18.8)
>90 days	137 (41.9)	30 (15.7) ***	30 (20.8) ***
Reason for referral, N (%)			
Dyspnea and other respiratory symptoms	120 (36.7)	75 (39.3)	47 (32.6)
Focal neurological symptoms	52 (15.9)	16 (8.4) *	16 (11.1)
Fall	38 (11.6)	17 (8.9)	17 (11.8)
Bleeding (all causes)	27 (8.3)	8 (4.2)	8 (5.6)
Altered consciousness	21 (6.4)	38 (19.9) ***	24 (16.7) ***
Discharge diagnosis, N (%)			
Infection	135 (41.3)	56 (29.3) **	
COVID-19		47 (24.6)	
Neurological	43 (13.1)	15 (7.9)	
Trauma	38 (11.6)	17 (8.9)	
Cardiac	23 (7.0)	9 (4.7)	
Cancer	20 (6.1)	7 (3.7)	
Hospitalization with death, N (%)	92 (28.1)	116 (60.7) ***	

* *p* < 0.05; ** *p* < 0.01; *** *p* < 0.001.

**Table 4 healthcare-12-01573-t004:** Correlation between NH health team composition and NH resident hospitalization in the last year of life in the pre-pandemic and COVID-19 pandemic periods. N—number.

	Prepandemic Period (N = 177)			COVID-19 Pandemic Period (N = 211)		
Variable	Last Year	Last 90 Days	Last 30 Days	Last Year	Last 90 Days	Last 30 Days
Physician/resident ratio	0.062	0.112	0.138	0.061	0.068	0.058
Nurse/resident ratio	0.228 **	0.275 **	0.330 **	0.131	0.129	0.262 **
Caregiver/resident ratio	0.065	0.143	0.159 *	0.087	0.072	0.101

Pearson correlation coefficient; * *p* < 0.05; ** *p* < 0.01.

## Data Availability

Data is contained within the article.

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
