# Peer review of "End-of-Life Care during the COVID-19 Pandemic: Decreased Hospitalization of Nursing Home Residents at the End of Life"

_healthcare, 2024, doi:10.3390/healthcare12161573_

Round 1
Reviewer 1 Report (Previous Reviewer 2)
Comments and Suggestions for Authors
The authors have responded appropriately to the suggested comments. However, they do not mention whether there are pharmacists in these NH, professionals who are extremely important and who can have a major impact on health outcomes and care. I consider this information to be highly important. Therefore, I believe that this information should be added to and completed with the information previously introduced about physicians, nurses and caregivers. If the units don't have pharmacists, the authors should mention this as a possible gap and present in the discussion how the presence or absence of these professionals can have an impact on the results obtained.
Comments on the Quality of English Language
Minor editing of English language required.
Author Response
The authors thank the reviewer for the analysis, and comments that contributed to improve the manuscript. We are pleased to address the reviewer comments and submit a revised version of our paper. Please find below a response to the comment:
Comment: The authors have responded appropriately to the suggested comments. However, they do not mention whether there are pharmacists in these NH, professionals who are extremely important and who can have a major impact on health outcomes and care. I consider this information to be highly important. Therefore, I believe that this information should be added to and completed with the information previously introduced about physicians, nurses and caregivers. If the units don't have pharmacists, the authors should mention this as a possible gap and present in the discussion how the presence or absence of these professionals can have an impact on the results obtained.
Thank you for bringing this to our attention. To address this issue additional information was added in the methods and conclusions sections, including new references:
Methods: “NH teams in the institutions enrolled include also social workers and therapists, however these professionals are not included in clinical decisions regarding hospitalizations, and therefore have not been included in the study. Clinical pharmacists, do not usually integrate the NH in Portugal, and were not a part of the NH care team in the studied institutions.” (lines 131-136)
Conclusion: “In Portugal NH remain officially a social care response to the elderly. The only mandatory health care professional is a nurse, in a limited period of the day. Even so, nowadays indoor medical support is the norm. However other health care professionals like clinical pharmacists, or nutritionists are very rarely included in the care teams of the NH [60]. Clinical pharmacists’ intervention in NH has been proved to be effective in ensuring quality use of medication, resulting in reduced fall rates [61], [62], [63]. Although evidence is less clear on the impact to reduce mortality or hospitalization and admission rates, filling this gap in the Portuguese NH might help to improve outcomes in end-of-life care[62].” (lines 337-344)
Reviewer 2 Report (New Reviewer)
Comments and Suggestions for Authors
The article addresses an important question and makes valuable contributions to the literature on end-of-life care during the COVID-19 pandemic. With a few improvements, the manuscript could become an important reference in the field.
Methods:
1. Page 3 (lines 119-120): provide detailed information on the interviews conducted: what questions were asked? who conducted the interviews? were the interviews recorded? how was the interview data analyzed?
2. Inform how the sample was stratified by region and justify this choice.
3. Instruments: Give more detail on the instruments used, in particular, on the psychometric properties of the instruments, where applicable.
4. A brief description of how the data was collected can help with the replicability of the study.
Discussion:
5. Discuss the implications of the results, especially in relation to end-of-life care practices and how these changes can be sustained after the pandemic.
6. Clarify the gains that the study's data brings to the scientific literature on the subject.
Author Response
The authors thank the reviewer for the thorough analysis, and the pertinent comments, that contribute to the improvement of the manuscript. We are pleased to address the reviewer comments and submit a revised version of our paper. Please find below a point-by-point response to the comments:
Comment 1: Page 3 (lines 119-120): provide detailed information on the interviews conducted: what questions were asked? who conducted the interviews? were the interviews recorded? how was the interview data analyzed?
Thank you for pointing this out. Further information was added: “A questionnaire covering the characteristics of the NH (number of residents, total of deaths in the institution in the study period, NH care team constitution and professional/ caregiver ratio) was previously sent to the participating NH. Data was collected in an interview conducted by the principal investigator with the NH management team” (lines 126-130)
Comment 2: Inform how the sample was stratified by region and justify this choice.
Thank you for allowing us to clarify this point. Further information was added: “A convenience sample was chosen to allow feasibility of the study. Stratification by region involved selecting a proportional number of participants in each Portuguese region, in order to improve representability of the data.” (lines 98-101)
Comment 3. Instruments: Give more detail on the instruments used, in particular, on the psychometric properties of the instruments, where applicable.
Thank you for your comment. The only instrument used in the study was ECOG PS. Additional data on inter-rater reliability was added: “The ECOG Performance Status Scale is a widely used method to assess the functional status of a patient. The instrument has an interrater reliability described in the literature of 0,69 to 0,83 (Spearmen rank correlation) [39].” (lines 117-119)
Comment 4: A brief description of how the data was collected can help with the replicability of the study.
Thank you for your comment. We rephrased the paragraph regarding this point: “Data was collected through analyses of the administrative and clinical records of the deceased residents, death certificates and hospital discharge information. “
Comment 5: Discuss the implications of the results, especially in relation to end-of-life care practices and how these changes can be sustained after the pandemic.
Comment 6. Clarify the gains that the study's data brings to the scientific literature on the subject
Thank you for allowing us to clarify these central issues of the manuscript. Given that the points are both generic, we considered most suited to address them jointly, and rephrased the entire conclusion section for better clarity: “This study brings new insights in the field of hospitalization at the end of life of NH residents during the COVID 19 pandemic. A significant decrease in hospitalizations was identified during the pandemic, although mortality increased (with excess mortality attributable to COVID-19 disease), and the rate of death at hospital remained stable. More residents died without visiting the emergency room in the last year of life and spent fewer nights at hospital. When hospitalization did occur, it was closer to death and with more serious clinical presentations. The decrease of hospitalizations was due mostly to potentially burdensome admissions, namely non-COVID-19 infections.
The deceased NH resident characteristics were similar in the period before and during the pandemic. Residents were very old, with high levels of dependency and multi-morbidity, with a possible ceiling effect reached before the pandemic.
A decrease in referrals, and less burdensome transitions suggest an improvement in end-of-life care practices during the pandemic period. This indicates that NH care teams were able to successfully meet some of the challenges of end-of-life care facing a devastating pandemic. The data suggest the influence of the role of physicians, nurses and caregivers in hospitalization of NH residents may be relatively independent of each other. The nursing team revealed to be particularly influential on transferal decisions. Improved training in end-of life care targeting the care team, prioritizing the nursing team, may contribute to sustain the positive changes described in the future. Further qualitative studies may highlight other factors contributing to this result and inform good practices that can be sustained in the future.”
Round 2
Reviewer 2 Report (New Reviewer)
Comments and Suggestions for Authors
No suggestions.
This manuscript is a resubmission of an earlier submission. The following is a list of the peer review reports and author responses from that submission.
Round 1
Reviewer 1 Report
Comments and Suggestions for Authors
The manuscript contains new and significant information to justify publication.
In general, the writing of the text as a whole has adequate spelling and verbal agreement. The authors address the topic with concise, precise, and clear language.
The title is consistent with the other items of the manuscript (objective, results, and conclusion) and provides important information about the main objective addressed by the study.
The summary presents sufficient information to understand the procedures performed and their outcome and is presented clearly and concisely. Conclusions relate to the objective.
The descriptors are coherent with the manuscript and will assist in identifying and increasing the visibility of this manuscript.
The Introduction is presented clearly and in a logical sequence. Explains the concepts used and the justification for the study. The study question was presented clearly.
The objective is related to the title, results, and conclusions.
The Method is consistent with the title and objective and provides a complete description of the methodological procedures. The authors do not comment on the use of the EQUATOR network framework, which has been recommended by numerous journals. Could the authors have justified the lack of description about the exclusion criteria, that is, were all deaths during the study period included? Was there no characteristic that could indicate bias in the results? The authors adequately present the instrument used and adequately explain the data analysis method.
The Results are consistent with the proposed objective. The discussion presents consistent interpretations compared to other studies with a similar theme.
The authors present a pertinent discussion and point out sensible conclusions that correspond to the results obtained and the proposed objective.
The conclusion is consistent with the development and findings of the study.
The references are current, and relevant and support the topic of the study.
I consider the manuscript relevant and with the potential to impact clinical practice.
Author Response
The authors thank the reviewer for the thorough analysis and the pertinent comments that contribute to improvement of the manuscript. We are pleased to address the reviewer comments and submit a revised version of our paper. Please find below a point-by-point response to the comments:
Comment 1: The authors do not comment on the use of the EQUATOR network framework, which has been recommended by numerous journals.
Thank you for bringing this to our attention. Additional information was added to the Study Design paragraph: “STROBE criteria were followed in the design and report of the present study (33)” (line 99-100).
Comment 2: Could the authors have justified the lack of description about the exclusion criteria, that is, were all deaths during the study period included? Was there no characteristic that could indicate bias in the results?
Thank you for pointing this out. Additional information was included and part of the paragraph on study population was rephrased to: “…All deaths that occurred within the study period were analysed. Death in the 90 days after admission was considered exclusion criteria, since the discussion of advanced directives and the relationship with the resident and their family may influence the decision to transfer the resident to acute care, and the place of death. A 90 day period to allow for these discussions to take place is recommended in the literature (34), (35)” (lines 111-117).
In the results section additional information was added: “A total of 454 NH deceased residents were screened for eligibility. A total of 67 were excluded due to death within the 90 days following admission.” (line 151-152).
Reviewer 2 Report
Comments and Suggestions for Authors
Although the topic is interesting and pertinent, I believe that the work presented needs some improvements. The inclusion of new information is essential if it is to be considered an effective asset for readers and for healthcare. The authors present the study and its results, but there is no in-depth discussion of the outcomes and applicability in clinical reality.
In order for this work to have the sufficient merit required to be published in this journal, Q2, I am attaching a document with a detailed presentation of comments that need to be answered and included in this work, so that it is eligible for publication.

English minor revision.
Author Response
The authors thank the reviewer for the thorough analysis, and the pertinent comments that contribute to the improvement of the manuscript. We are pleased to address the reviewer comments and submit a revised version of our paper. Please find below a point-by-point response to the comments:
Comment 1 (line 13): This should include Materials.
We agree. Correction inserted as suggested.
Comment 2 (line 15): This information should be part of the materials and not the results.
We agree. Correction according to the suggestion.
Comment 3 (line 21): Authors should specify which preventable causes.
Thank you for pointing this out. Sentence rephrased to “ … due to potentially burdensome hospitalizations.” For better clarity. Also, in the conclusions section the same correction was introduced: (line 323-325) “The decrease of hospitalizations was due mostly to potentially burdensome admissions, namely non-COVID-19 infections.”
Comment 4 (line 99): Here, the authors should characterize the NHs in terms of the team. Detail the constitution of the teams, because this is a very relevant aspect for healthcare today. Then, in the results, the authors should include a section covering the association between the constitution of the teams and the results in terms of hospitalizations, deaths, length of hospital stay,..... This is a crucial aspect for the value of this work. This information objectively allows this work to contribute to the restructuring of NHs in terms of teams.
We agree with the reviewer, this is a major and very important topic, needed to inform care provision and policy making. Bearing this in mind, this topic was addressed in a previous article, that aimed to identify and characterize acute care transitions in the last year of life of NH residents and identify its predictors. NH care team constitution was one of the potential predictors studied, and interesting results were described and discussed. In the present study, we intended to focus specifically on the impact of the covid pandemic on the decision to hospitalize NH residents at the end of life. To emphasize the importance of this crucial aspect, additional information was added: “The NH team characterization and its impact on NH resident hospitalization during the pre-pandemic period has been previously described[26]. The constitution of the care team remained stable before and during the pandemic. No external team support was provided during the pandemic to the NH enrolled in the study.” (lines 135-139)
Comment 5 (line 294): I consider that, before the conclusion, it is essential that the authors include a section on concrete future prospects. That is, based on the results obtained in this study, what specific measures do they suggest should be implemented in NHs to improve residents' health care, particularly in specific end-of-life care practices. It is essential that the authors discuss the need for a permanent multidisciplinary team in NHs, including doctors, pharmacists, nurses, assistants, nutritionists, psychologists, etc.
Thank you for this suggestion. An additional section was included (lines 326-342):
“Future prospects:
The results obtained can be interpreted as an improvement of end-of-life care practices in the NH during the pandemic. The NH teams were able to provide adequate care to residents with infection in the last year of life, and avoid potentially burdensome hospitalisations [45]. Kasdorf and col. described as the most relevant aspects that can aid in reducing unnecessary transitions: timely identification and communication of the last year of life; consideration of palliative care options; availability and accessibility of care services; and having a healthcare professional taking main responsibility for care planning [7]. During the pandemic public discussion on ceilings of care and identification of prognosis factors might have contributed to the awareness, both by the care team and the residents and their families, about the need to provide proportional care to frail elderly patients [22], [59]. The multidisciplinary care team at the NH was able to better identify the needs of the residents and its own capacities for symptom control, and optimize care delivery, using a more proportional approach. Fostering multidisciplinary care team practices and training at the NH, involving multiple health and social care professionals (including doctors, nurses, assistants, pharmacists, nutritionists, psychologists, social workers), might contribute to improve end-of-care practices [60].”
Reviewer 3 Report
Comments and Suggestions for Authors
Thank you for presenting your work, overall the article is well presented. Please see specific line-by-line editing suggestions included below:
Lines 36-39- This sentence is a bit convoluted, consider rephrasing to simplify the sentence and remove the first clause.
Line 41 - Consider changing ‘dissemination’ to ‘high transmission’ to utilize disease state terminology.
Line 39-42- It may be helpful for the reader’s comprehension of the topic if a very brief explanation or example of what it meant by the organization and the architectural structure of Nursing homes and how that contributed to the spread of COVID.
Line 45-46- The description of ‘the symptom management could be complex, 45 due to the nature of the disease’ seems to imply that the topic is referring to end of life care specific to those experiencing COVID. If the intention is to refer to all individuals end of life care regardless of COVID status, consider revising this portion of the sentence for clarity.
Line 50- Consider changing ‘face’ to ‘manage’ for clarity
Line 55- Portugal is introduced and then not mentioned again in the purpose of the article, please provide additional context for the study being conducted specific to Portugal.
Lines 59-61- Consider rewriting and combining these sentences for clarity and flow.
Line 66- Add a comma after “pandemic”
Line 68- Consider changing ‘in the NH’ to ‘of NH residents’
Line 72- Consider changing ‘better’ to ‘increased’
Lines 72-74- Who are you referring to when mentioning increased knowledge of options- the nursing home staff, residents, residents families, all of the above? Please add a little more clarity here about the impact of increasing knowledge of options and who could benefit from it. Another suggestion that this should be moved to the discussion or conclusion section instead of having it in the introduction since this is a possible application of the knowledge gained in this study, not a component of the current data collection.
Line 97- Change ‘its’ to ‘their’
Line 137- Change ‘by’ to ‘of’
Lines 137-138- Old age is a subjective characteristic, please consider defining this statement more concretely.
Line 173- Change ‘was’ to ‘were’
Line 205- Please describe what you mean by ‘excess’ to improve the reader's understanding of the topic.
Line 216- Remove the commas after ‘studies’ and after ‘admissions’
Lines 246-247- Consider rewording the second half of this sentence to improve clarity and flow.
Line 252- Change ‘did not suffer a significant change’ to ‘did not change significantly’
Line 293- Consider using a term other than ‘significant’ here since it is not referring to the outcomes of data analysis.
Line 299- Change ‘less’ to ‘fewer’
Comments on the Quality of English LanguageI found only minor edits that need to be made to improve syntax. These are included in the line by line feedback.
Author Response
The authors thank the reviewer for the thorough analysis, and the pertinent comments that contribute to the improvement of the manuscript. We are pleased to address the reviewer comments and submit a revised version of our paper. Please find below a point-by-point response to the comments.
Comment 1 (lines 36-39) - This sentence is a bit convoluted, consider rephrasing to simplify the sentence and remove the first clause.
Thank you for pointing this out. The paragraph has been rephrased for better clarity: “The COVID-19 pandemic posed unprecedented challenges to the care of NH residents. It was caused by a new coronavirus (2019-nCoV), first identified as causing an acute respiratory syndrome in humans in November 2019, in Wuhan, China [10]. The disease rapidly evolved to a pandemic declared by the World Health Organization in March 2020 [11]. Knowledge on the pathogenesis, clinical expression and adequate preventive and treatment measures was built during the ongoing the pandemic. Amid scientific uncertainty, WHO, and country guidance and regulations have often been contradictory in the early stages of the pandemic. Also, personal protective equipment, testing and later vaccination has suffered severe shortages[11], [12], [13], [14]….” (lines 38-49).
Comment 2 (line 41) - Consider changing ‘dissemination’ to ‘high transmission’ to utilize disease state terminology.
Agree. Correction inserted as suggested (line 52).
Comment 3 (line 39-42) - It may be helpful for the reader’s comprehension of the topic if a very brief explanation or example of what it meant by the organization and the architectural structure of Nursing homes and how that contributed to the spread of COVID.
Thank you for bringing this to our attention. Additional information added: “Given its organization, architectural characteristics (closed systems, proximity between users and staff), and susceptible population (age, comorbidities, specific care needs) [15], …” (line 50-52).
Comment 4 (line 45-46) - The description of ‘the symptom management could be complex, 45 due to the nature of the disease’ seems to imply that the topic is referring to end of life care specific to those experiencing COVID. If the intention is to refer to all individuals end of life care regardless of COVID status, consider revising this portion of the sentence for clarity.
Thank you for pointing this out. The intention was to address COVID-19 disease. Sentence was rephrased to “… the symptom management could be complex, due to the nature of the COVID-19 disease…” for improved clarity (line 57).
Comment 5 (line 50) - Consider changing ‘face’ to ‘manage’ for clarity.
Agree. Correction inserted as suggested (line 61).
Comment 6 (line 55) - Portugal is introduced and then not mentioned again in the purpose of the article, please provide additional context for the study being conducted specific to Portugal.
Thank you for bringing this to our attention. Sentence was rephrased for better clarity: “Also in Portugal, NH residents are frequently referred to the emergency department in the last year of life, and death at hospital is frequent, in both the general population and the NH residents [27], [28]” (lines 67-69).
Additional information was added: “In the Portuguese context a decrease in general hospital admissions has been described, but the NH resident has not been previously studied [34].” (lines 81-83).
Comment 7 (lines 59-61) - Consider rewriting and combining these sentences for clarity and flow.
Thank you for the suggestion. Sentence was rephrased as recommended: “During the pandemic a decrease in hospital admissions was identified both in the general population [29], [30] and in NH residents[31], [32], [33].” (lines 70-73).
Comment 8 (line 66) - Add a comma after “pandemic”
Agree. Correction inserted as suggested (line 78).
Comment 9 (line 68) - Consider changing ‘in the NH’ to ‘of NH residents’
Agree. Correction inserted as suggested (line 79-80).
Comment 10 (line 72) - Consider changing ‘better’ to ‘increased’
After considering comment 11, all the paragraph was rephrased.
Comment 11 (lines 72-74) - Who are you referring to when mentioning increased knowledge of options- the nursing home staff, residents, residents families, all of the above? Please add a little more clarity here about the impact of increasing knowledge of options and who could benefit from it. Another suggestion that this should be moved to the discussion or conclusion section instead of having it in the introduction since this is a possible application of the knowledge gained in this study, not a component of the current data collection.
Following the pertinent suggestion, and combining this comment with a reviewer 2 comment, this topic was moved to the discussion and rephrased: “The results obtained can be interpreted as an improve of end-of-life care practices in the NH during the pandemic. The NH teams were able to provide adequate care to residents with infection in the last year of life, and avoid potentially burdensome hospitalisations [45]. Kasdorf and col. described as the most relevant aspects that can aid in reducing unnecessary transitions: timely identification and communication of the last year of life; consideration of palliative care options; availability and accessibility of care services; and having a healthcare professional taking main responsibility for care planning [7]. During the pandemic public discussion on ceilings of care and identification of prognosis factors might have contributed to the awareness, both by the care team and the residents and their families, about the need to provide proportional care to frail elderly patients [22], [59]. The multidisciplinary care team at the NH was able to better identify the needs of the residents and its own capacities for symptom control, and optimize care delivery, using a more proportional approach. Fostering multidisciplinary care team practices and training at the NH, involving multiple health and social care professionals (including doctors, nurses, assistants, pharmacists, nutritionists, psychologists, social workers), might contribute to improve end-of-care practices [60].” (lines 327-342)
Comment 12 (line 97)- Change ‘its’ to ‘their’
Agree. Correction inserted as suggested (line 113).
Comment 13 (line 137) - Change ‘by’ to ‘of’
Agree. Correction inserted as suggested (line 162).
Comment 14 (lines 137-138) - Old age is a subjective characteristic, please consider defining this statement more concretely
Thank you for pointing this out. Sentence was rephrased to “residents died aged 86.1±8.7 years” (line 162-163).
Comment 15 (line 173) - Change ‘was’ to ‘were’
Agree. Correction inserted as suggested (line 198).
Comment 16 (line 205) - Please describe what you mean by ‘excess’ to improve the reader's understanding of the topic.
Thank you for the suggestion. Rephrased to “ … with an increase in the number of deaths above what would normally be expected based on the experience from previous years” (lines 231-232).
Comment 17 (line 216) - Remove the commas after ‘studies’ and after ‘admissions’ .
Agree. Correction inserted as suggested (line 242).
Comment 18 (lines 246-247) - Consider rewording the second half of this sentence to improve clarity and flow.
Thank you for the suggestion. The second half was related to the rest of the paragraph and could not be removed, however, to improve clarity and flow, the first half of the sentence was replaced at the end of the paragraph “Death from infectious causes other than COVID-19 decreased significantly during the pandemic period. This phenomenon has been described in other studies, with the lower mortality from other transmissible diseases being attributed to measures taken to mitigate the spread of COVID-19 infection, namely stricter hygiene standards, regular disinfection, social distancing and screening of visitors [13], [15]. Death from noninfectious causes remained stable during the pandemic. (lines 273-279)
Comment 19 (line 252) - Change ‘did not suffer a significant change’ to ‘did not change significantly’.
Agree. Correction inserted as suggested (line 280-281).
Comment 20 (line 293) - Consider using a term other than ‘significant’ here since it is not referring to the outcomes of data analysis.
Thank you for pointing this out, “significant” was removed, it was an unnecessary adjective (line 323).
Comment 21 (line 299) - Change ‘less’ to ‘fewer’
Agree. Correction inserted as suggested (line 348).
Round 2
Reviewer 2 Report
Comments and Suggestions for Authors
Although the authors refer to a previous study to characterize the health team at NHs, the aforementioned study does not present the impact of health professionals on outcomes in detail. The idea would be to identify the teams in each institution and try to establish correlations with the results obtained. For example, in NHs that have a full-time pharmacist, were the results better? From my point of view, this analysis is also essential for the discussion and conclusions of this article.
The authors have not carried out this proposed analysis, so I leave it up to the editor to make the final decision on publication.
Comments on the Quality of English LanguageMinor editing of English language required.
